developmental biology, evolution, physiology

avian brood parasites, co-evolutionary arms race, embryonic development, muscle development

**Author for correspondence:**
Stephanie C. McClelland
e-mail: stephanie.mcclelland.2018@live.rhul.ac.uk

# Embryo movement is more frequent in avian brood parasites than birds with parental reproductive strategies

Stephanie C. McClelland[1], Miranda Reynolds[1], Molly Cordall[1], Mark E. Hauber[2,3], Wolfgang Goymann[4,5], Luke A. McClean[7], Silky Hamama[8], Jess Lund[6,7], Tanmay Dixit[6], Matthew I. M. Louder[2], Ignas Safari[4,5,9], Marcel Honza[10], Claire N. Spottiswoode[6,7] and Steven J. Portugal[1]

[1]Department of Biological Sciences, School of Life and Environmental Sciences, Royal Holloway University of London, Egham, Surrey TW20 0EX, UK
[2]Department of Evolution, Ecology, and Behavior, School of Integrative Biology, University of Illinois, Urbana-Champaign, IL 61801, USA
[3]American Museum of Natural History, New York, NY 10024, USA
[4]Max-Planck-Institut für Ornithologie, Abteilung für Verhaltensneurobiologie, Eberhard-Gwinner-Str. 6a, D-82319 Seewiesen, Germany
[5]Coucal Project, PO Box 26, Chimala, Tanzania
[6]Department of Zoology, University of Cambridge, Downing Street, Cambridge CB2 3EJ, UK
[7]FitzPatrick Institute of African Ornithology, DST-NRF Centre of Excellence, University of Cape Town, Rondebosch 7701, Cape Town, South Africa
[8]c/o Musumanene Farm, PO Box 630038, Choma, Zambia
[9]Department of Biology, University of Dodoma, PO Box 338, Dodoma, Tanzania
[10]The Czech Academy of Sciences, Institute of Vertebrate Biology, Květná 8, 603 65 Brno, Czech Republic

SCM, 0000-0002-8763-2291; MEH, 0000-0003-2014-4928; WG, 0000-0002-7553-5910; TD, 0000-0001-5604-7965; MIML, 0000-0003-4421-541X; IS, 0000-0001-5157-5398; SJP, 0000-0002-2438-2352

Movement of the embryo is essential for musculoskeletal development in vertebrates, yet little is known about whether, and why, species vary. Avian brood parasites exhibit feats of strength in early life as adaptations to exploit the hosts that rear them. We hypothesized that an increase in embryonic movement could allow brood parasites to develop the required musculature for these demands. We measured embryo movement across incubation for multiple brood-parasitic and non-parasitic bird species. Using a phylogenetically controlled analysis, we found that brood parasites exhibited significantly increased muscular movement during incubation compared to non-parasites. This suggests that increased embryo movement may facilitate the development of the stronger musculoskeletal system required for the demanding tasks undertaken by young brood parasites.

## 1. Introduction

Movement is essential for successful embryonic development across vertebrates [1,2]. Embryonic movement shapes the development of an animal's musculo-skeletal system and ranges from sporadic twitching of muscle tissue in the early stages of development, to coordinated motions akin to walking or flying closer to hatching [3,4]. While embryonic movement has been acknowledged as vital in embryogenesis and growth, most research focus has been on identifying and mitigating the molecular causes of low movement. Little attention has been given to understanding how and why movement affects the embryo's form and function, despite evidence that movement can affect phenotypic expression [3]. Paralysing chick embryos, for example, causes malformation of joints, reduced muscle tone and stunted bone growth [1,5,6].

Conversely, experimentally manipulated hyperactivity increases the density of primary muscle fibres in chick embryos, which is a key factor determining the potential for post-natal muscle growth [2]. As with adult animals, embryonic 'exercise' causes the muscle to become stronger and larger [7,8]. This presents a possible mechanism by which animals that require exceptional muscular strength in early life might achieve the necessary musculature.

One such group of animals is the avian brood parasites. Obligate avian brood parasites are birds that lay their eggs in the nests of other species (hosts), forcing them to raise the parasitic offspring [9]. This strategy requires specialized physiological and behavioural adaptations in the eggs and young to survive in the host nest [10,11]. Some of these adaptations could be shaped by embryonic movement. For example, shorter incubation periods and stronger eggshells have independently arisen across multiple brood parasite lineages [11,12], and greater strength and stamina are required to hatch from these stronger structures [13,14]. The musculature for this task must be developed in the relatively short ontogenetic period within the egg [9]. Additionally, to fledge successfully, many brood-parasitic young must ensure that they receive most, or all, of the food provisioned by the foster parents [15,16], by either out-competing or killing the host young [9,17,18]. These strategies are physically strenuous and require a level of strength, coordination and energy expenditure that is not usually seen in altricial offspring (i.e. species that hatch in an underdeveloped state and are reliant on direct parental care). Embryo movement could, therefore, provide mechanical stimulation for the development of a stronger musculoskeletal system to support a parasitic lifestyle, and result in the convergent acquisition of higher rates of embryonic movement in distantly related parasitic species.

Here, we tested the hypothesis that increased embryonic movement assists avian brood parasites to achieve the necessary muscular and skeletal development needed for both the tasks of hatching from thicker eggshells and, in highly virulent species, killing or out-competing their nest-mates. We measured the rate of embryonic movement over the course of incubation across a range of brood parasites, their hosts and their non-parasitic relatives. This allows us to test the prediction that avian brood parasites should exhibit a higher embryonic movement rate (EMR) relative to closely related non-parasitic species.

While most brood parasites tend to hatch from stronger eggshells than other species, other aspects of their early life physical demands differ between brood-parasitic species, and this variation is largely associated with their level of virulence (defined by [19]). The chicks of highly virulent parasites remove or destroy host eggs or chicks [19,20], whereas less virulent brood-parasitic species use physical size advantage and exaggerated begging behaviours to outcompete host nest-mates and receive sufficient provisioning from the host parents [19,21]. Eviction of host young, a strategy used by many highly virulent parasitic species, likely imposes a significant strain on the skeleton of the newly hatched parasite chick, and this could potentially cause skeletal damage if not compensated by increased muscular support, or denser or more ossified bones [22,23]. Evidence of increased musculature in a virulent brood parasite has been observed in the chicks of common cuckoos (*Cuculus canorus*), which have a higher density of muscle fibres in their *musculus complexus*,

the hatching muscle in their necks, compared to non-parasitic birds [24]. This is speculated to be an adaptation for hatching from significantly thicker eggshells, but may also facilitate the eviction of host eggs and chicks. Given the evidence that muscle development is shaped by embryonic activity in birds, the increased embryonic movement provides a plausible mechanism by which denser and stronger muscles, including the *musculus complexus*, could be developed by young common cuckoos and other parasitic species.

This range of behaviours exhibited by parasitic chicks inspires predictions about differences in embryonic movement among parasite species. Specifically, if increased embryonic movement increases the strength capabilities of hatchlings, then highly virulent parasitic species—i.e. those which require greater physical exertion to eject or kill host young—should show a further increase in their rate of embryo movement compared to less virulent parasitic species. However, the muscular demands of less virulent parasitic species should be greater than those of non-parasitic species, since less virulent species must still outcompete host young, typically through heightened begging.

## 2. Results

Using a non-invasive method to measure embryonic muscle twitching, we recorded EMR as the number of embryo movements per minute, repeatedly measured over the period of incubation, in 437 eggs from 14 species of birds, including five host–parasite systems from three continents. The incubation period was divided into five stages to standardize embryonic development (electronic supplementary material, figure S1 and table S1), and egg size was accounted for in the analyses. While egg size improved the fit of the model, it did not significantly predict EMR (see statistical methods). After controlling for phylogenetic relatedness (figure 1), we found that brood parasites had a significantly higher overall rate of increase in EMR over the course of incubation (slope of interaction between parasite status and incubation stage) compared to non-parasitic species (phylogenetically controlled mixed model (PMM), slope ± s.e. = 7.28 ± 1.85, $t$ = 3.94, $p$ = 0.002; figure 2). Phylogeny explained a small percentage of the observed variance in EMR ($H^2$ = 0.17 ± 0.09), indicating that EMR is not strongly predicted by species position within the phylogeny (i.e. species relatedness). This supports the hypothesis that reproductive strategy (parasitic versus parental) is the main determinant of EMR over the course of incubation, as opposed to phylogenetic relatedness. Across all species, EMR significantly increased with incubation stage (PMM, estimate ± s.e. = 16.11 ± 1.08, $t$ = 14.89, $p$ < 0.001; figure 2).

When we compared individual species pairs of hosts and parasites, linear mixed models (LMMs) showed differences between most brood parasite species and their hosts, in the rate of increase in EMR over the incubation period. For instance, common cuckoos had a significantly greater increase in EMR across incubation compared to their hosts, great reed warblers (*Acrocephalus arundinaceus*) (slope ± s.e. = −7.03 ± 2.66, $t_{832}$ = −2.64, $p$ = 0.008; figure 3a), and also compared to one of the two non-parasitic cuckoo species recorded, white-browed coucals (*Centropus superciliosus*) (slope ± s.e. = 12.36 ± 5.64, $t_{735}$ = −2.19, $p$ = 0.03; figure 3a), but not the other, African black coucals (*Centropus grillii*)

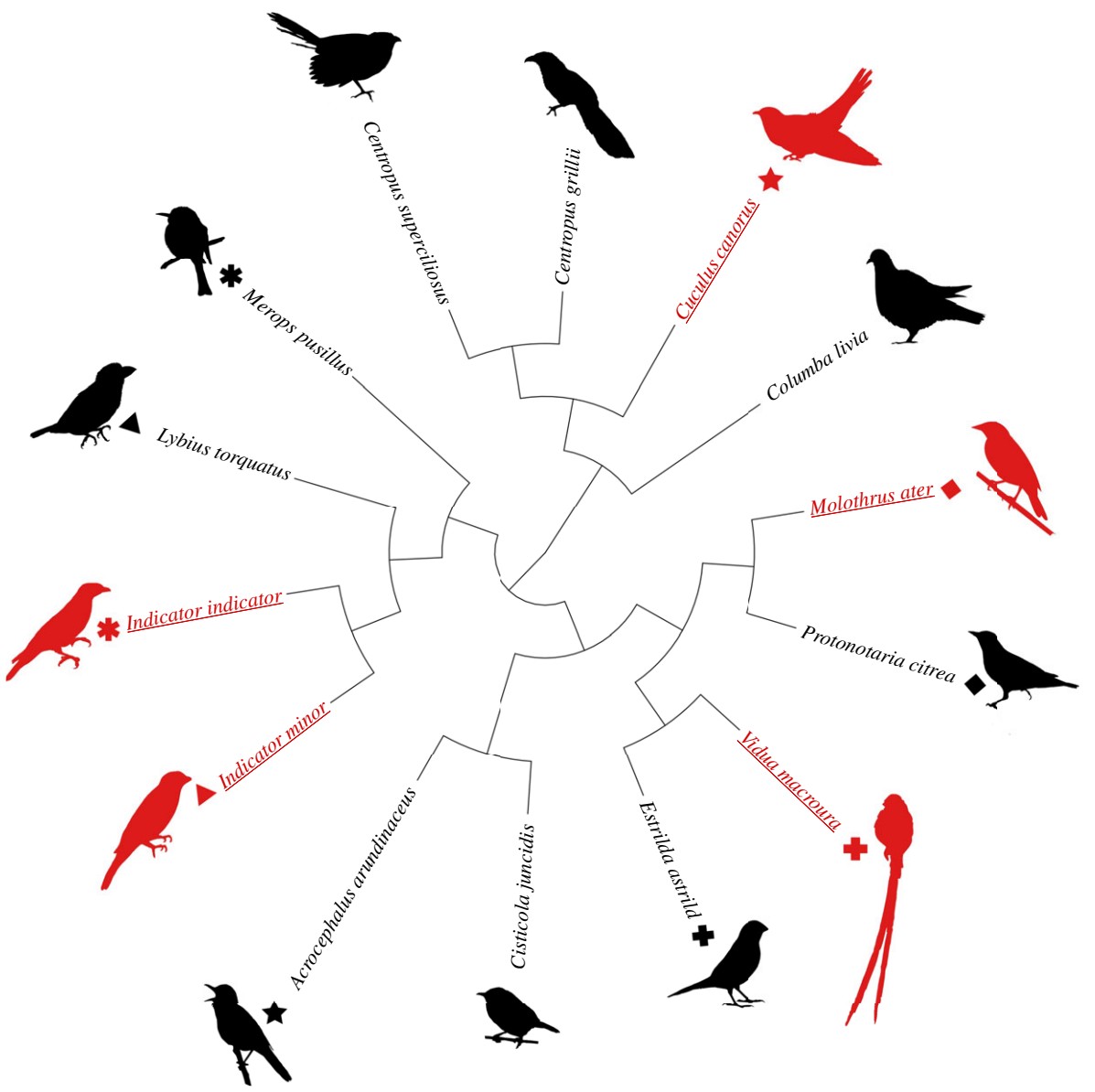

**Figure 1.** Phylogenetic tree showing the species in the PMM. Symbol shapes match brood parasites (underlined and red in online version) to the host species (black, not underlined) that they parasitize. Constructed from the 'Tree of Life' database using the R package 'rotl' [13,14]. Branch lengths set at 1. (Online version in colour.)

(slope ± s.e. = 9.25 ± 5.98, $t_{765}$ = 1.55, $p$ = 0.12; figure 3a). This suggests that the demands of hatching and virulence in common cuckoos may have driven their relatively high EMR.

Similarly, lesser honeyguides (*Indicator minor*) increased their EMR over incubation at a significantly higher rate than their hosts, black-collared barbets (*Lybius torquatus*) (LMM, slope ± s.e. = 15.36 ± 7.10, $t_{189}$ = 2.16, $p$ = 0.03; figure 3b). The increase in EMR of lesser honeyguides was also significantly higher than that of the congeneric, greater honeyguides (*Indicator indicator*) (slope ± s.e. = 17.81 ± 8.67, $t_{182}$ = 2.05, $p$ = 0.041; figure 3b). Unlike the lesser honeyguides and their hosts, the slope of increase of EMR in greater honeyguides did not differ significantly from that of their hosts, little bee-eaters (*Merops pusillus*) (slope ± s.e. = 2.91 ± 7.67, $t_{191}$ = 0.38, $p$ = 0.70; figure 3b), which themselves had a relatively high EMR.

Of the two low virulent parasites measured, brown-headed cowbirds (*Molothrus ater*) exhibited a significantly steeper slope of increase of EMR over incubation than their hosts, prothonotary warblers (*Protonotaria citrea*) (LMM, slope ± s.e. = −12.95 ± 5.90, $t_{88}$ = 2.20, $p$ = 0.03; figure 3c). However, stage 1 cowbirds also had an EMR that was

lower than correspondingly aged prothonotary warbler embryos, resulting in the steep slope of increase seen in cowbird eggs (table 1). The other low virulence species, pin-tailed whydahs (*Vidua macroura*), did not significantly differ in the slope of EMR increase compared to their hosts, common waxbills (*Estrilda astrild*) (LMM, slope ± s.e. = 10.90 ± 8.11 $t_{99}$ = 1.34, $p$ = 0.18; figure 3d). Overall, among parasitic species, we did not find a significant difference between high virulence and low virulence species (LMM, slope ± s.e. = 6.29 ± 4.26, $t_{486}$ = 1.48, $p$ = 0.14; table 1 indicates which parasite species are categorized as high or low virulence). The mean EMR of each species of parasite and host at each stage of incubation is shown in table 1.

## 3. Discussion

Brood-parasitic species displayed a significantly higher over-all rate of embryonic muscle movement over the course of incubation, compared to both their host species and to other closely related non-parasitic species. There was also interspecific variation in the rates of increase in embryonic

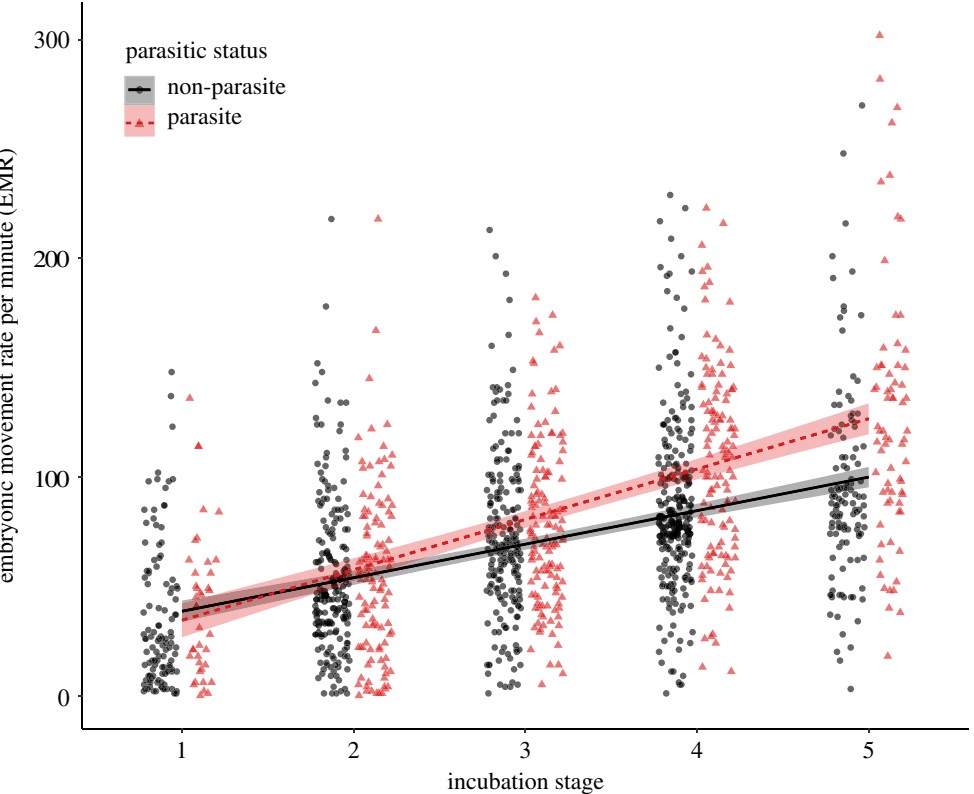

**Figure 2.** Rate of embryo movement per minute (EMR) over the course of incubation for all parasitic species (triangles and dashed line, red in online version) and all non-parasitic species (black and solid line) combined. Incubation stages 1 to 5 are described in the electronic supplementary material, figure S1. Shading indicates standard errors. (Online version in colour.)

movement over the incubation period, with the EMR of most brood-parasitic species increasing at a significantly steeper rate as incubation progressed, compared to their hosts or closely related non-parasitic species. The steeper slope of increase in parasites meant that the differences were particularly evident in the later stages of their incubation, where brood parasites exhibited especially high EMRs. In particular, common cuckoos, lesser honeyguides and brown-headed cowbirds demonstrated exceptionally high rates of embryo movement near the end of their incubation period. These findings are consistent with our hypothesis that embryonic movement may evolve in response to selection for the demands of brood parasitism on both the embryo before hatching, and on the newly hatched chick.

Brood parasitism imposes selection for greater strength both before hatching and shortly after hatching. Increased embryonic movement in brood parasites prior to hatching is consistent with our hypothesis that embryonic movement may facilitate developing the muscular strength required to hatch from exceptionally strong eggs. A steeper increase in embryonic movement throughout incubation and, particularly during late incubation, could facilitate the development of the musculature and stamina required to break out of a stronger eggshell [7]. As thicker eggshells have been shown to be common across brood parasites regardless of virulence [25], the lack of difference seen between highly virulent and less virulent species might be explained by strong selection for hatching ability, which may overshadow selection for the post-hatching demands of these species. Upon hatching, other demands of brood parasitism (e.g. out-competing nest-mates, or killing or evicting host chicks) are unlikely to be mutually exclusive, as

similar muscle development could be required for these tasks. Many muscle complexes have multiple functions for which the rate of development could be optimized. The musculus complexus in the neck of birds is important for the process of hatching and has been shown to be enlarged in the necks of common cuckoos [26]. However, this muscle complex is also important for begging behaviour as it regulates dorsal flexion of the neck and coordination of head movement [27] and so affects competitive interactions between nestlings. This may give an advantage to nest-mate-tolerant parasite species, such as brown-headed cowbirds, which are known to beg more intensely than non-parasitic species [28,29].

We did not find any consistent difference in the slope of increase of EMR between high virulence and low virulence species of brood parasites, contrary to our second prediction that muscular demands of virulent species would require greater embryonic movement than less virulent parasite strategies. The lack of a correlation between EMR and virulence may be due to considerable interspecific variation in parasitic virulence strategies and, therefore, variable selection on musculature. For example, the natural history of virulence differs between the two species of honeyguides we studied, despite their phylogenetic closeness. Both species kill host nest-mates shortly after hatching by biting and shaking them vigorously [30]; however, the demands of killing host young differ greatly between greater and lesser honeyguides. Greater honeyguide females puncture host eggs when they lay their own such that few host eggs hatch, reducing the demands on the parasite chick to remove their competition [30,31]. Adult lesser honeyguides do not puncture host eggs, meaning that lesser honeyguide young must themselves kill the full brood of host young of up to four chicks [32]. Moreover,

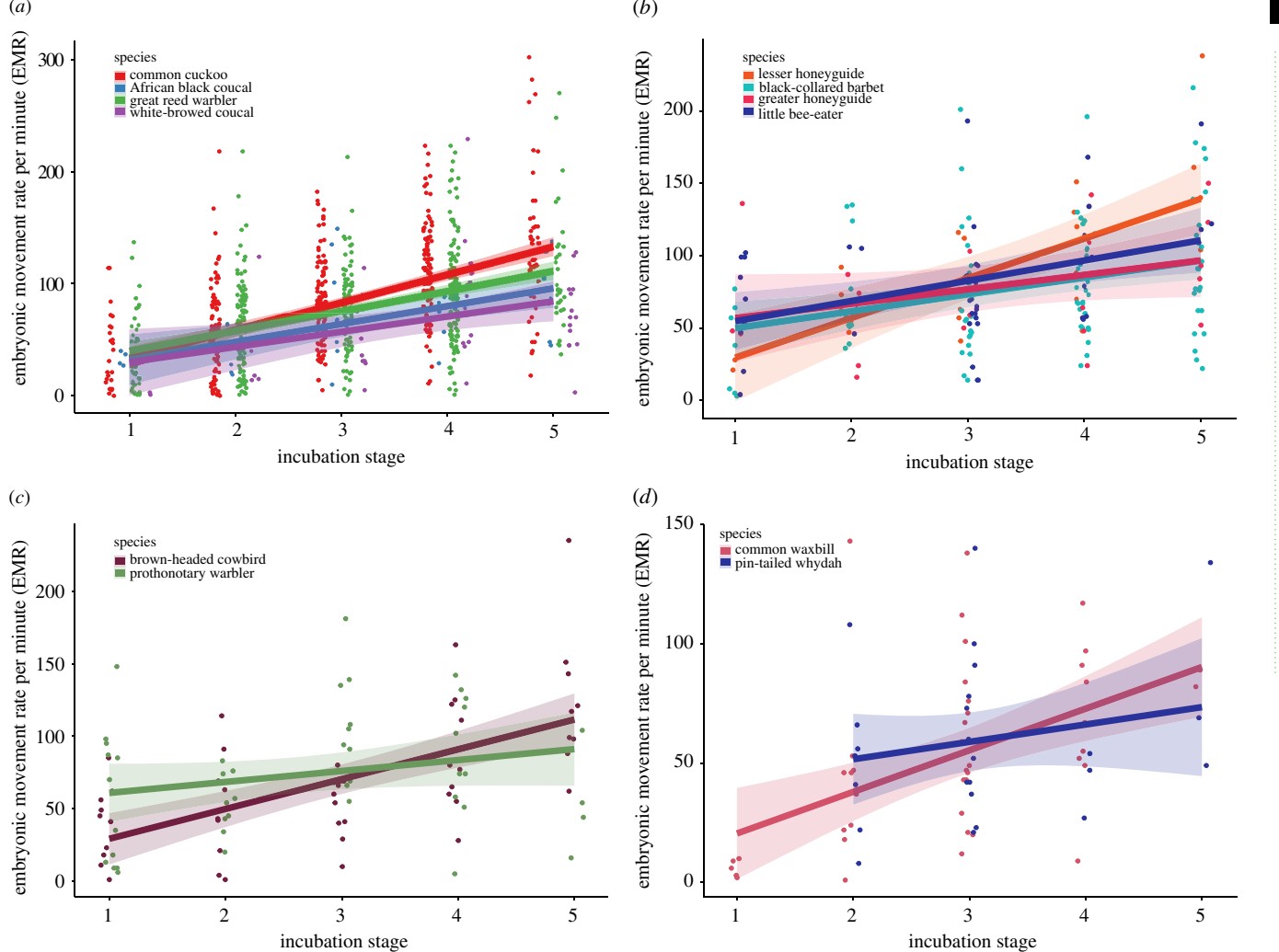

**Figure 3.** Rate of embryo movement per minute (EMR) over the course of incubation for (a) common cuckoos, their great reed warbler hosts, and two non-parasitic cuckoos: African black coucals and white-browed coucals, (b) lesser honeyguides and their hosts (black-collared barbets) and greater honeyguides and their hosts (little bee-eaters), (c) brown-headed cowbirds and their hosts, prothonotary warblers and (d) pin-tailed whydahs and their hosts, common waxbills. (No measurements were available for pin-tailed whydahs at stage 1.) Shading indicates standard errors.

the chicks of greater honeyguides are larger than the chicks of their respective hosts, while lesser honeyguides parasitize black-collared barbets, whose nestlings are approximately twice the mass of a lesser honeyguide chick [33]. This may explain why lesser honeyguides had significantly higher rates of embryonic movement than greater honeyguides, and why greater honeyguides did not have higher rates than their hosts. That these two honeyguide species are congeneric [34] makes this difference particularly striking, as it suggests that embryonic behaviour has the potential to evolve rapidly in response to differences in host behaviour and morphology. Additionally, little bee-eaters, hosts of the greater honeyguides, exhibited relatively high rates of embryo movement compared to other non-parasite species, a curious finding for which we currently do not have an explanation.

We propose that increased EMR is a shared characteristic in the embryonic development of brood parasites that has evolved convergently between lineages. The factors influencing variation in embryonic movement across incubation are less well understood, as are the mechanisms that control this movement. A potential factor which could facilitate higher embryonic movement in parasites could be the thermal properties of parasite eggs, which have been shown to

retain heat for longer periods during incubation breaks due to their thicker shell [35]. This could influence any potential temperature-mediated activity of the embryo. Hormones in the egg may play a role in regulating embryo movement. There is evidence that maternally deposited androgens in the egg effect the embryonic growth and early life behaviour of birds, although their role in embryonic movement has not been studied, to our knowledge [36,37]. The phylogenetic position did not significantly predict a species' EMR, suggesting that variation in EMR is driven primarily by intrinsic or environmental factors rather than common ancestry; however, an extensive genetic study would be required to determine the genes involved or whether embryonic movement constitutes an epigenetic source of variation on embryo development, depending on how embryonic movement is regulated [1,38]. The five species of brood parasites we studied represent four of the seven known evolutionary origins of brood parasitism in birds [39,40], suggesting that this is potentially an embryonic adaptation to a brood-parasitic lifestyle that has evolved convergently in independent brood-parasitic lineages, as has been proposed for other physiological traits [11,41].

The behaviour of brood-parasitic hatchlings is extraordinary and demonstrates their exceptional physical abilities.

**Table 1.** Mean rate of embryo movement (EMR) per minute and standard errors at each incubation stage (1–5), for parasitic species and their hosts. Parasites are in italics. Designation of high virulence or low virulence of parasite species based on [19].

| species | stage 1 (EMR, mean ± s.e.) | stage 2 (EMR, mean ± s.e.) | stage 3 (EMR, mean ± s.e.) | stage 4 (EMR, mean ± s.e.) | stage 5 (EMR, mean ± s.e.) |
|---|---|---|---|---|---|
| *common cuckoos* (high virulence) | 39 ± 6.7 | 55.9 ± 4.2 | 83.0 ± 3.8 | 111.6 ± 4.4 | 129.8 ± 9.0 |
| great reed warblers | 35.1 ± 4.4 | 61.5 ± 4.2 | 76.3 ± 4.1 | 92.5 ± 4.3 | 98.1 ± 9.4 |
| *lesser honeyguides* (high virulence) | 24.5 ± 3.5 | 65.8 ± 7.8 | 82 ± 13.1 | 101.2 ± 15.9 | 148 ± 25.4 |
| black-collared barbets | 40.5 ± 9.4 | 73.8 ± 11.5 | 73.8 ± 8.43 | 81.3 ± 6.9 | 98.0 ± 10.5 |
| *greater honeyguides* (high virulence) | 76.6 ± 29.7 | 52.6 ± 13.9 | 70.7 ± 16.4 | 88.3 ± 16.8 | 100.4 ± 16.8 |
| little bee-eaters | 64.2 ± 12 | 74.0 ± 10.3 | 70.7 ± 9.1 | 97.8 ± 13.0 | 135.0 ± 18.9 |
| *brown-headed cowbirds* (low virulence) | 39.1 ± 7.8 | 49.8 ± 12.8 | 53.3 ± 6.3 | 88.6 ± 12.8 | 123.8 ± 16.9 |
| prothonotary warblers | 56.1 ± 13.6 | 54.0 ± 7.0 | 104.3 ± 12.2 | 88.4 ± 13.8 | 54.5 ± 18.3 |
| *pin-tailed whydahs* (low virulence) | NA | 50.0 ± 25.6 | 62.2 ± 12.1 | 42.7 ± 8.1 | 84.0 ± 25.6 |
| common waxbills | 6.0 ± 1.5 | 43.7 ± 12.1 | 59.9 ± 8.3 | 69.0 ± 10.7 | 81.3 ± 4.6 |

Here, we have shown that the behaviour of the embryo during development could shape the physiology of brood parasites, and so may be a key factor in the successful exploitation of their hosts. While the evidence is consistent with our theory of an adaptive function of embryo movement, this cannot be conclusively determined with these data, and it is possible that other aspects of a parasitic lifestyle could induce or select for greater movement. Future research could address this by directly measuring the consequences of greater EMRs on the muscle density and performance of individuals of these parasitic species. For example, if this embryonic trait has an adaptive benefit for brood parasites, then we would expect increased embryo movement to increase the parasitic chick's efficiency at evicting or killing host offspring. Additionally, it would be informative to measure embryo movement in other species that experience challenging nestling social environments, such as nest-sharing colonial breeders or species with large asynchronous clutches and/or high rates of siblicide [42,43]. Further to their relevance for brood-parasitic species, our findings suggest that embryo movement may be a generally overlooked process in the evolution of the diverse life histories, forms and behaviours observed in birds.

## 4. Methods

### (a) Embryo movement quantification

EMR was measured using a portable digital egg monitor ('Egg Buddy™', Avitronic Services, Abbotskerswell, Devon, UK). The use of the Egg Buddy for biological research was validated by [44], and it has been used to monitor embryo development and heart rate in both birds and reptiles [45–47]. To quantify the frequency of EMR, the egg is placed on a rubber cup inside the egg monitor chamber. The monitor transmits a beam of infrared light through the egg and detects any disruption to the beam caused either by movement of the embryo, or the contraction of blood vessels in response to a heartbeat [44]. Embryo movement is reliably detected in altricial embryos after approximately the first quarter of the incubation period [46]. However, as all eggs in this study were compared to each other at the same incubation stages (see staging description below), any reduced accuracy in earlier incubation would not influence comparisons. In early incubation, heart rate is detectable before muscle twitching becomes evident. However, as the size and activity of the embryo increases, heart rate measurement becomes more challenging to record due to the increased muscular movements of the embryo [44]. Therefore, we did not record heart rate.

Each subject egg was placed into the chamber of the monitor immediately after removal from the nest or incubator and allowed to acclimatize in the darkened interior for approximately 30 s. Longer acclimation periods were not performed to prevent the egg from excessive cooling. The egg was positioned with the long axis of the egg roughly perpendicular to the laser beam, with slight adjustments made to the angle if the movement was not initially detected. If no movement was detected after this, no measurement was made to minimize disturbance. Embryo movement was displayed in real-time on the screen of the egg monitor as an animated bird symbol which changed configuration when movement was detected, and a 60 s video of the screen was recorded immediately following acclimation and subsequently analysed. The number of embryo movements was counted from watching the video recordings at 0.5× speed. This was performed by either SCM, MC or MR, and blindly to the specimen ID.

### (b) General field methods overview

Nests were monitored *in situ* at several field locations (detailed below). Nests of the host or focal (non-parasitic relatives of parasites) species were located and visited frequently during the early egg-laying stage to detect brood parasitism. Eggs were marked with a pencil or felt-tip marker upon completion of the clutch, for later identification. When a nest was parasitized, it was not

disturbed for the first 2 days of incubation so as not to interfere with natural egg rejection or acceptance by host parents. Eggs were visited from the second or third day of incubation, depending on species, and measurements of embryo movement were taken for the parasite egg and then a randomly selected host egg. Where possible we measured only a single egg per host clutch to avoid pseudo-replication since host eggs in the same clutch are non-independent. However, due to the limited nest availability of zitting cisticolas and little bee-eaters, two eggs were sampled from the same clutch. This was accounted for statistically by including nest identity as a random factor in all analyses. The measurements were taken close to the nest to minimize the time that eggs were out of the nest, and eggs were out of the nest no longer than 10 min in total. The same host and parasite eggs were then measured again every second day until hatching. Repeat measures were not obtained for some eggs due to clutch loss from predation, nest destruction or host rejection. The feasibility of estimating exact incubation start dates varied with species and field site, and therefore sometimes an estimate of embryo age by candling the egg and assigning a stage system was required [48–50] (electronic supplementary material, figure S1 and table S1). Egg stage was estimated by visual examination of the embryo via candling and a stage of development assigned based on the embryo's size and appearance, the albumen coloration, blood vessel quantity and air cell size. Further details and illustrations of embryo stages are available in the electronic supplementary material, figure S1.

## (c) Field sites and study species
### (i) Zambia (dry season)
Data were collected on greater honeyguides (*Indicator indicator*) and their hosts, little bee-eaters (*Merops pusillus*), and lesser honeyguides (*Indicator minor*) and their hosts, black-collared barbets (*Lybius torquatus*), at a field site (16°45′ S, 26°54′ E) on farms in the Choma District of Zambia during the dry season (September to November) of 2016, 2017 and 2018. For further details of the field site, see [51]. Nests were found by local field assistants. Black-collared barbet nests were located in tree cavities and accessed by openings that were cut into the cavity wall above the nest and covered by strips of bark between visits. The nests of little bee-eaters were located in underground tunnels dug into the side of the burrows of aardvarks (*Orycteropus afer*). These nests could be accessed by digging down to the nest from the ground above, as described in [30].

Nests were visited and embryo movement recordings taken every 2–3 days during the incubation period. Nests were often located after the beginning of incubation, and incubation stage (electronic supplementary material, figure S1) was estimated by candling the egg and assessing embryo development. Exact incubation day was unknown for these nests. Embryo movement measurements were taken from the parasite egg in each nest located, along with a live host egg if present. Greater honeyguide females often puncture the host eggs when they lay their own [51], and so most parasitized nests of little bee-eaters did not contain live host eggs; therefore, measurements were also taken from non-parasitized little bee-eater nests. We were unable to obtain measurements from greater honeyguides during early incubation (prior to incubation stage 2).

### (ii) Zambia (wet season)
Data were collected from eggs of pin-tailed whydahs (*Vidua macroura*), and of its hosts, common waxbills (*Estrilda astrild*). Additional data were collected from zitting cisticolas (*Cisticola juncidis*), which are common hosts of the cuckoo finches (*Anomalospiza imberbis*). However, low parasitism rates during the 2019 and 2020 breeding season meant insufficient data were collected on this parasite to include it in this study. Data from zitting

cisticolas were included as a non-parasitic species for phylogenetic comparison.

These data were gathered at the same field site described above during the wet season (February to March) of 2019 and 2020. Both common waxbills and zitting cisticolas build nests close to the ground in grassy habitat. Nests were found by local field assistants and were measured every 2 days. Due to differences in the length of incubation between species, the incubation period from onset of incubation until hatching for each species was divided into five stages, from stage 1 to stage 5 (electronic supplementary material, figure S1). Incubation stage was estimated by candling, and incubation commencement and hatching date were known for most nests (electronic supplementary material, figure S1). No measurements were made for pintailed whydahs at stage 1 due to difficulty locating nests and low parasitism rates during these years.

### (iii) Czech Republic
Data were collected from common cuckoos (*Cuculus canorus*) parasitizing great reed warblers (*Acrocephalus arundinaceus*). The nests of great reed warblers were located in narrow strips of reed beds surrounding ponds in the south of the Czech Republic (48°54′ N, 16°59′ E). Parasitized nests were visited either every day or every 2 days during incubation and eggs were briefly removed and brought to the bank of the pond for measurement. The eggs were replaced with decoys while measurements were taken and returned to the nest within 10 min. Abandoned cuckoo eggs were transferred to the laboratory and incubated until hatching. For details about the incubation procedure, see [26]. Measurements were taken from these eggs also. Measurements from incubator-hatched eggs ($n = 18$ of 68) and wild-hatched eggs were not statistically different in EMR ($t_{41} = 0.906$, $p = 0.366$) so these data were combined for analysis. The chicks which hatched from these eggs were returned to other nests at the field site.

### (iv) Illinois, USA
Measurements were collected on brown-headed cowbirds (*Molothrus ater*) from a nest-box breeding population of their hosts, prothonotary warblers (*Protonotaria citrea*) (for further details see [52]) during the summer of 2018. Nest-boxes were sited on the edges of a swamp on public land in Illinois, USA (37°24′ N, 88°53′ W) and had high rates of parasitism during the study year (approx. 80%). The egg monitor was set up on dry land close to the nest-box and eggs were removed for less than 10 min for measurements. For nests that contained two cowbird eggs, both parasitic eggs were measured since they should be laid by different females. Eggs were measured every 2 days across incubation.

### (v) Tanzania
Measurements were collected on socially polyandrous African black coucals (*Centropus grillii*) and socially monogamous white-browed coucals (*C. superciliosus*) in the Usangu wetland in south-western Tanzania (8°41′ S, 34°5′ E). Coucals build dome-shaped nests in dense vegetation. These nests were located either by observing birds carrying nesting material or incubating birds back to the nest, or by following birds equipped with radio-transmitters (for further details see [53,54]). The egg monitor was set up *ca* 5–10 m from the nest, and eggs were removed for less than 10 min for measurements. Eggs were measured every 4 days during incubation.

### (vi) United Kingdom
Measurements were taken on the eggs of domestic homing pigeons (*Columba livia*) at Royal Holloway University of London.

Pigeons nested in purpose-built housing lofts (2.1 × 1.8 m) on the campus of Royal Holloway University of London, and recordings were taken at the lofts on alternate days between incubation days 3 and 20. Husbandry details are available in [55].

## (d) Statistical methods

All statistical analyses were conducted in R [56] using 'R Studio' [57]. EMR was defined as the number of movements per minute recorded by the egg monitor. Measurements at stage 1 that recorded 0 EMR were excluded from analysis, as false zeros were possible due to the small size of the embryo.

We used phylogenetically controlled analyses for our comparison of EMR between these 14 species, as species cannot be considered statistically independent due to shared ancestry [58,59]. The inclusion of two species of non-parasitic cuckoos (white-browed coucals and black coucals) provided within-group phylogenetic control for common cuckoos, as the latter are more distantly related to their hosts than the other paired-species (host–parasite) in these analyses [60,61]. The honeyguides (Indicatoridae) are a sister group to the barbets (Lybidae) which are hosts to lesser honeyguides, and hence this host provided a suitable comparison. The phylogenetic relatedness of our focal species was constructed and downloaded from the Open Tree of Life and using the 'rotl' package [14] in R v. 3.3.2 (figure 1a). Using this phylogenetic tree, we constructed PMM [62] to compare the rate of EMR per stage between all species using the package 'sommer' R v. 4.0 [63]. The phylogenetic element of this model allowed us to separate the percentage of variance in EMR that is potentially explained by phylogeny, from any variance that could be attributed to parasitic lifestyle, or other life-history factors. The phylogenetic signal of the trait (EMR) was calculated as the percentage of variance explained by phylogeny as a proportion of the total variance in EMR and is presented as $H^2$. This value is comparable to Pagel's lambda in other analyses [64]. The 'emtrends' function using the package 'emmeans' R v. 1.4.6 [65] was applied to the PMM to compare the slope of increase in EMR over incubation stage in parasites and non-parasites.

PMMs were constructed with EMR as response variation and a combination of incubation stage, parasitic status, fresh egg mass, breeding latitude and mean incubation length as predictor variables. Akaike's information criterion (AIC) scores of these models were then compared to determine the best-fitting model to explain the data, where the best-fitting model was at least 2 AIC points lower than the next lowest AIC. Neither mean incubation length nor mean breeding latitude of species (values taken from [66]) were retained in the final model as neither were statistically significant and did not improve the fit of the model by greater than 2ΔAIC. Egg mass significantly improved the fit of the model by more than 2 AIC points and was retained in the final model, but was not statistically significant (1.17 ± 0.94, $t = -1.24$, $p = 0.26$). Egg identity was included as a random variable in all models to account for repeated measurements from the same egg at different incubation stages. Similarly, nest identity was included as a random variable to account for eggs that were sampled from the same host nest. The model with the best fit for predicting EMR in these species included fresh egg mass and the interaction of parasitic status and incubation stage as fixed factors, and egg identity and nest identity as random factors.

Species-to-species comparisons were also undertaken using separate LMM (using the lmer function in the package 'lmerTest') [67] to examine potential differences between each parasite species and their respective hosts. Common cuckoos, great reed warblers and both coucal species were compared in a single LMM, and *post hoc* testing was used to compare species to each other. As with the prior analyses, egg and nest identity were included as random effects to account for repeated measurements from the same eggs or clutch. Species identity and the interaction between parasitic status and incubation stage were included as predictor variables. As with the phylogenetic models, species breeding latitude was not found to be a significant or informative predictor for EMR and was therefore dropped from the final model.

Data accessibility. All data are available in the electronic supplementary material [68].

Authors' contributions. S.C.M.: conceptualization, data curation, formal analysis, funding acquisition, investigation, methodology and writing-original draft; M.R.: data curation, investigation and methodology; M.C.: data curation, investigation, writing-review and editing; W.G.: data curation, investigation, resources, writing-review and editing; L.M.: investigation, writing-review and editing; S.H.: investigation and resources; J.L.: investigation, resources, writing-review and editing; M.H.: conceptualization, investigation and resources; M.E.H.: conceptualization, resources, writing-review and editing; T.D.: investigation, resources, writing-review and editing; M.I.M.L.: investigation and resources; I.S.: investigation and resources; C.S.: conceptualization, resources, supervision, writing-review and editing; S.P.: conceptualization, investigation, resources, supervision, writing-review and editing All authors gave final approval for publication and agreed to be held accountable for the work performed therein.

Competing interests. Authors declare no competing interests.

Funding. S.C.M. was supported by a London NERC DTP Studentship; S.J.P. was funded by a Research Project (grant no. RPG-2018-332) from the Leverhulme Trust, Czech fieldwork was supported partially by the Czech science foundation (GR CR) project no. S 17-12262S, and C.N.S. and Zambian fieldwork were supported partially by a BBSRC David Phillips Fellowship (BB/J014109/1) and by the DST-NRF Centre of Excellence at the FitzPatrick Institute, University of Cape Town. M.I.M.L. and M.E.H. were supported by the USA National Science Foundation and the Wissenschaftskolleg zu Berlin. W.G. was supported by the Max-Planck-Gesellschaft; I.S. was funded by a scholarship from the Ministry of Education and Vocational Training (MoEVT) Tanzania, the German Academic Exchange Service (DAAD) and the International Max Planck Research School (IMPRS) for Organismal Biology.

Acknowledgements. We thank everyone who assisted us during fieldwork at multiple field sites, including Jack Thirkell, Gabriel Jamie, Jessie Walton, Poyo Makomba, Musa Makomba, Collins Moya, Michel Šulc, Milica Požgayová, Petr Procházka, Jeffrey Hoover and Wendy Schelsky. We thank Lackson Chama, Moses Chibesa and Stanford Siachoono at Copperbelt University for their support. We thank Richard and Vicki Duckett, Troy and Elizabeth Nicolle, and Ian and Emma Bruce-Miller for permission to work on their farms, and Molly and Archie Greenshields for providing us with a home during fieldwork in Zambia. We thank the many people who helped us find nests in Zambia, particularly Lazaro Hamusikili, Tom Hamusikili, Sanigo Mwanza, Sylvester Munkonko and Calisto Shankwasiya. We thank the Department of National Parks and Wildlife in Zambia, Tanzania Wildlife Research Institute (TAWIRI), and the Tanzanian Commission for Science and Technology (COSTECH) for support and permits.

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
