## [Peer Review File · Proceedings of the Royal Society B: Biological Sciences]

Review History

RSPB-2021-1137.R0 (Original submission)

Review form: Reviewer 1

Recommendation

Accept with minor revision (please list in comments)

Scientific importance: Is the manuscript an original and important contribution to its field?

Excellent

General interest: Is the paper of sufficient general interest?

Excellent

Quality of the paper: Is the overall quality of the paper suitable?

Excellent

Is the length of the paper justified?

Yes

Should the paper be seen by a specialist statistical reviewer?

No

Do you have any concerns about statistical analyses in this paper? If so, please specify them explicitly in your report.

No

It is a condition of publication that authors make their supporting data, code and materials available - either as supplementary material or hosted in an external repository. Please rate, if applicable, the supporting data on the following criteria.

Is it accessible?

N/A

Is it clear?

N/A

Is it adequate?

N/A

Do you have any ethical concerns with this paper?

No

Comments to the Author

This is an exciting study that explores the idea that embryo movement in birds is shaped by selection. Using a phylogenetically controlled analysis, the authors find a pattern of more embryo movement in brood parasites than their hosts. The interpretation is that higher embryonic movement rate in brood parasites evolved convergently between lineages because a) brood parasites engage in activities that require strength (breaking out of thick egg shell, evicting host hatchlings), and b) brood parasites with a stronger musculoskeletal system would be more successful at both activities. Embryo movement is predicted to strengthen muscular development, though this was not tested. Your description of data scoring from the buddy egg monitor needs clarification and I suggest a different reference for its use in birds.

Comments

Line 157: "This suggesting" is awkward

Table 1: you show that little bee-eaters have a lot of embryo movement; somewhere in the manuscript text or table you may want to indicate that little bee eaters excavate sand tunnels. Therefore, embryo movement in general could be selected for in systems with high activity level, such as excavation behaviour. Perhaps such a thought could be included as a footnote for this table as the embryo movement pattern in little bee-eaters is strikingly different from that recorded in other non-brood parasites.

Lines 249-270: Your explanations are convincing; however, see comment above regarding behavioural ecology of the host, which you neglect in this paragraph.

Line 269: embryonic movement does not evolve. This needs to be rewritten.

Line 275: change "with" to "that"

Line 285: change from "rate is driven primarily by ecology rather than common ancestry" to "rate is driven primarily by intrinsic or environmental factors rather than common ancestry"

Discussion in general: you do not use the opportunity to make predictions for future research. For example, the magnitude of embryonic movement could predict time to evict host and a range of other measurable parameters.

Line 309: replace reference 44 with Colombelli-Négrel, D., Hauber, M.E. and Kleindorfer, S., 2014. Prenatal learning in an Australian songbird: habituation and individual discrimination in superb fairy-wren embryos. *Proceedings of the Royal Society B: Biological Sciences*, 281(1797), p.20141154.

Colombelli-Négrel et al (2014) were the first to use the “Egg Buddy” for avian research, the same model you also used – previous research in birds used acoustocardiography. The study you cite, reference 44 published in 2018, followed on from Colombelli-Négrel et al (2014).

Line 318: the heart rate measure does NOT become unreliable; rather, there is more movement. The egg buddy clearly shows the difference between heart rate bpm and movement on the screen with different symbols (heart symbol, moving bird symbol). You could still have recorded/analysed heart rate but you would need to restrict the notation of heart rate to the seconds without embryo movement.

Line 328: you write “embryo movement was displayed in real time on the screen of the egg monitor” and then line 330: “The number of embryo movements were counted from playback of the video recording”. This is confusing and needs to be rewritten to reflect the fact that the buddy egg monitor screen has a symbol showing embryo movement per second; the display duration on the screen is 1 sec for both heart rate as beats per minute and for any movement event even if the event was less than 1 sec); so what you can score is the number of seconds that had embryo movement. Also, to avoid confusion with the term “playback the video” (because a common research approach is to use playback to elicit a behavioural response) you could state that you watched the video and scored the number of seconds with embryo movement events.

Lines 344-346: you write that you sampled one host egg per nest, and each egg was out of the nest for max 10 minutes. This does not need to be repeated elsewhere.

Line 372: this contradicts one host egg per nest when you write “along with any live host eggs present”

Line 410: why did you return hatched cuckoo chicks to unparasitised host nests? You collected the abandoned cuckoo egg from the wild; why would you return an abandoned cuckoo egg that you reared in a lab to a new host nest?

Line 419-420: repeats the info about one host egg per nest

Lines 442-443: the egg monitor records seconds with egg movement; this needs to be rewritten. Perhaps you included seconds with both ‘bird movement symbol’ and ‘zero’ as movement. If so, this needs to be stated. A bird movement symbol would never be a false zero. A zero would indicate that no signal could be detected (movement or heart rate). I agree that intense movement could result in a zero signal, and this is unlikely in very young embryos.

Review form: Reviewer 2

Recommendation

Major revision is needed (please make suggestions in comments)

Scientific importance: Is the manuscript an original and important contribution to its field?

Good

General interest: Is the paper of sufficient general interest?

Acceptable

Quality of the paper: Is the overall quality of the paper suitable?

Acceptable

Is the length of the paper justified?

Yes

Should the paper be seen by a specialist statistical reviewer?

No

Do you have any concerns about statistical analyses in this paper? If so, please specify them explicitly in your report.

No

It is a condition of publication that authors make their supporting data, code and materials available - either as supplementary material or hosted in an external repository. Please rate, if applicable, the supporting data on the following criteria.

Is it accessible?

Yes

Is it clear?

Yes

Is it adequate?

Yes

Do you have any ethical concerns with this paper?

No

Comments to the Author

This study was aimed to test the hypothesis that (lines 92-95) increased movement during the embryonic life would allow avian brood parasites to achieve the necessary muscular development to successfully exploit their hosts (e.g., hatching from thicker eggshells, killing or out-competing their non-parasitic nestmates). The authors found that embryos of some avian brood parasites exhibit increased muscular movement compared to some of their main host species, and some closely-related non-parasitic species. On the other hand, there seems to be no link between embryonic movement rate and parasite virulence. I enjoyed reading the manuscript. Overall, this is an interesting and well-written manuscript that provide novel results on a little explored aspect of avian brood parasitism. However, I have some concerns regarding the main hypothesis of the study and the interpretation of results.

1) The main hypothesis of the study focuses on the potential adaptive benefits that increased embryonic movement would have for brood parasitic species (this idea is present throughout the entire manuscript); however, the authors do not provide any evidence of such benefits (e.g., correlation between embryonic movement rate (EMR) and brood parasite's muscle development; a link between EMR and hatching success or competitive abilities). I know it is difficult to fix at this point, but author should provide some direct evidence on the competitive advantage (benefit) that increased embryonic movement would provide for brood parasites. Otherwise, one should be more cautious when referring to the increased EMR as "embryonic adaptation".

2) Similarly, some predictions would need to be supported by stronger evidence. For example, the authors should be more cautious in speculating on the muscular demands, and the degree of musculoskeletal development, required by the different parasitic strategies (lines 121-126). Why are virulent brood parasites (those that eliminate all host eggs or kill the host offspring) expected to have higher muscular demands compared to less virulent brood parasites (those that share the

nest with the parasitic offspring and, therefore, have to compete with their nestmates)? Is there any evidence to support that idea? If not, would ejector parasites be necessarily predicted to show increased embryonic movement rate than non-ejector parasites?

3) Interpretation of results. This study shows that some avian brood parasites have a higher overall rate of increase in EMR over the course of incubation compared to non-parasitic species. While this is an interesting result, the authors should be cautious in stating that this is an “embryonic adaptation to a brood-parasitic lifestyle [...] explained through convergent evolution”. To state that, they should provide evidence for the potential adaptive benefit of increased EMR. Do the authors have any data (e.g., hatching success, begging rate) to suggest that brood parasites benefit from increased embryonic movement? In my opinion, this would make the difference and would improve the study substantially.

Other comments:

Lines 272-274: Please be more cautious in these statements. While results are interesting, they do not show that an increased embryonic movement rate has adaptive benefits for brood parasites.

Lines 313-314: “Embryo movement is reliably detected in altricial embryos from approximately 25% into the incubation” This sentence would need to be reworded/clarified.

Line 315: “the same incubation stage”. Considering that the study includes different bird species – some of them relatively distant phylogenetically –, the concept of “incubation stage” should be used carefully when comparing different species. I'm just saying that it should be carefully explained in the main text (not just in the supplementary information), especially considering that it is a key concept in this study.

Lines 319-320: “heart rate”. Was heart rate not recorded or not included? Please, make it clear if heart rate and EMR were measured separately. It should be clearly stated if heart rate was decided not to include, or if data on heart rate are being neglected because muscle movement prevents to quantify it.

Lines 327-328: “If no movement was detected after this, no measurement was made”. Why? Please clarify it.

Line 344: Why was pseudoreplication avoided considering only host eggs? Parasitic and host eggs in the same nest are also non-independent.

Line 349: “or other naturally occurring factors”. Such as which? Please specify such “naturally occurring factors” or remove this sentence.

Lines 391-394: Incubation stages should be described earlier (in “General field methods”) and in much more detail.

Decision letter (RSPB-2021-1137.R0)

02-Jul-2021

Dear Ms McClelland:

Your manuscript has now been peer reviewed and the reviews have been assessed by an Associate Editor. The reviewers' comments (not including confidential comments to the Editor) and the comments from the Associate Editor are included at the end of this email for your

reference. As you will see, the reviewers and the Editors have raised some concerns with your manuscript and we would like to invite you to revise your manuscript to address them.

Research ethics:

Use of animals and field studies:

It is a condition of publication that you make available the data and research materials supporting the results in the article. Please see our Data Sharing Policies (<https://royalsociety.org/journals/authors/author-guidelines/#data>). Datasets should be deposited in an appropriate publicly available repository and details of the associated accession number, link or DOI to the datasets must be included in the Data Accessibility section of the article (<https://royalsociety.org/journals/ethics-policies/data-sharing-mining/>). Reference(s) to datasets should also be included in the reference list of the article with DOIs (where available).

Please submit a copy of your revised paper within three weeks. If we do not hear from you within this time your manuscript will be rejected. If you are unable to meet this deadline please let us know as soon as possible, as we may be able to grant a short extension.

Best wishes,
Dr Sasha Dall
mailto:proceedingsb@royalsociety.org

Associate Editor

Comments to Author:

I have received two expert reviews of the manuscript. Both reviewers and I found this to be a very interesting study, tackling an interesting research question in an innovative way. The study uses phylogenetically-controlled analyses to compare embryonic movement in brood-parasitic and non-parasitic birds, and finds that brood-parasitic species have increased movement throughout embryonic development, suggesting convergent evolution. Both reviewers provide a list of suggestions that I hope the authors will find helpful in improving the manuscript. Reviewer 2 raises an important point about non-independence of parasitic and host eggs, and this needs addressing. Both reviewers point out that evidence for the benefits of embryonic movement in post-embryonic life is not provided, and I agree that additional discussion of this would strengthen the paper. Both reviewers also ask for clarification of the methods.

Reviewer(s)' Comments to Author:

Referee: 1

Comments to the Author(s)

This is an exciting study that explores the idea that embryo movement in birds is shaped by selection. Using a phylogenetically controlled analysis, the authors find a pattern of more embryo movement in brood parasites than their hosts. The interpretation is that higher embryonic movement rate in brood parasites evolved convergently between lineages because a) brood parasites engage in activities that require strength (breaking out of thick egg shell, evicting host hatchlings), and b) brood parasites with a stronger musculoskeletal system would be more successful at both activities. Embryo movement is predicted to strengthen muscular development, though this was not tested. Your description of data scoring from the buddy egg monitor needs clarification and I suggest a different reference for its use in birds.

Comments

Line 157: "This suggesting" is awkward

Table 1: you show that little bee-eaters have a lot of embryo movement; somewhere in the manuscript text or table you may want to indicate that little bee eaters excavate sand tunnels. Therefore, embryo movement in general could be selected for in systems with high activity level, such as excavation behaviour. Perhaps such a thought could be included as a footnote for this table as the embryo movement pattern in little bee-eaters is strikingly different from that recorded in other non-brood parasites.

Lines 249-270: Your explanations are convincing; however, see comment above regarding behavioural ecology of the host, which you neglect in this paragraph.

Line 269: embryonic movement does not evolve. This needs to be rewritten.

Line 275: change "with" to "that"

Line 285: change from "rate is driven primarily by ecology rather than common ancestry" to "rate is driven primarily by intrinsic or environmental factors rather than common ancestry"

Discussion in general: you do not use the opportunity to make predictions for future research. For example, the magnitude of embryonic movement could predict time to evict host and a range of other measurable parameters.

Line 309: replace reference 44 with Colombelli-Négrel, D., Hauber, M.E. and Kleindorfer, S., 2014. Prenatal learning in an Australian songbird: habituation and individual discrimination in superb fairy-wren embryos. *Proceedings of the Royal Society B: Biological Sciences*, 281(1797), p.20141154.

Colombelli-Négrel et al (2014) were the first to use the "Egg Buddy" for avian research, the same model you also used – previous research in birds used acoustocardiography. The study you cite, reference 44 published in 2018, followed on from Colombelli-Négrel et al (2014).

Line 318: the heart rate measure does NOT become unreliable; rather, there is more movement. The egg buddy clearly shows the difference between heart rate bpm and movement on the screen with different symbols (heart symbol, moving bird symbol). You could still have recorded/analysed heart rate but you would need to restrict the notation of heart rate to the seconds without embryo movement.

Line 328: you write "embryo movement was displayed in real time on the screen of the egg monitor" and then line 330: "The number of embryo movements were counted from playback of the video recording". This is confusing and needs to be rewritten to reflect the fact that the buddy egg monitor screen has a symbol showing embryo movement per second; the display duration on the screen is 1 sec for both heart rate as beats per minute and for any movement event even if the event was less than 1 sec); so what you can score is the number of seconds that had embryo movement. Also, to avoid confusion with the term "playback the video" (because a common research approach is to use playback to elicit a behavioural response) you could state that you watched the video and scored the number of seconds with embryo movement events.

Lines 344-346: you write that you sampled one host egg per nest, and each egg was out of the nest for max 10 minutes. This does not need to be repeated elsewhere.

Line 372: this contradicts one host egg per nest when you write "along with any live host eggs present"

Line 410: why did you return hatched cuckoo chicks to unparasitised host nests? You collected the abandoned cuckoo egg from the wild; why would you return an abandoned cuckoo egg that you reared in a lab to a new host nest?

Line 419-420: repeats the info about one host egg per nest

Lines 442-443: the egg monitor records seconds with egg movement; this needs to be rewritten. Perhaps you included seconds with both 'bird movement symbol' and 'zero' as movement. If so, this needs to be stated. A bird movement symbol would never be a false zero. A zero would indicate that no signal could be detected (movement or heart rate). I agree that intense movement could result in a zero signal, and this is unlikely in very young embryos.

Referee: 2

Comments to the Author(s)

This study was aimed to test the hypothesis that (lines 92-95) increased movement during the embryonic life would allow avian brood parasites to achieve the necessary muscular development to successfully exploit their hosts (e.g., hatching from thicker eggshells, killing or out-competing their non-parasitic nestmates). The authors found that embryos of some avian brood parasites exhibit increased muscular movement compared to some of their main host species, and some closely-related non-parasitic species. On the other hand, there seems to be no link between embryonic movement rate and parasite virulence. I enjoyed reading the manuscript. Overall, this is an interesting and well-written manuscript that provide novel results on a little explored aspect of avian brood parasitism. However, I have some concerns regarding the main hypothesis of the study and the interpretation of results.

1) The main hypothesis of the study focuses on the potential adaptive benefits that increased embryonic movement would have for brood parasitic species (this idea is present throughout the entire manuscript); however, the authors do not provide any evidence of such benefits (e.g., correlation between embryonic movement rate (EMR) and brood parasite's muscle development; a link between EMR and hatching success or competitive abilities). I know it is difficult to fix at this point, but author should provide some direct evidence on the competitive advantage (benefit) that increased embryonic movement would provide for brood parasites. Otherwise, one should be more cautious when referring to the increased EMR as "embryonic adaptation".

2) Similarly, some predictions would need to be supported by stronger evidence. For example, the authors should be more cautious in speculating on the muscular demands, and the degree of musculoskeletal development, required by the different parasitic strategies (lines 121-126). Why are virulent brood parasites (those that eliminate all host eggs or kill the host offspring) expected to have higher muscular demands compared to less virulent brood parasites (those that share the nest with the parasitic offspring and, therefore, have to compete with their nestmates)? Is there any evidence to support that idea? If not, would ejector parasites be necessarily predicted to show increased embryonic movement rate than non-ejector parasites?

3) Interpretation of results. This study shows that some avian brood parasites have a higher overall rate of increase in EMR over the course of incubation compared to non-parasitic species. While this is an interesting result, the authors should be cautious in stating that this is an "embryonic adaptation to a brood-parasitic lifestyle [...] explained through convergent evolution". To state that, they should provide evidence for the potential adaptive benefit of increased EMR. Do the authors have any data (e.g., hatching success, begging rate) to suggest that brood parasites benefit from increased embryonic movement? In my opinion, this would make the difference and would improve the study substantially.

Other comments:

Lines 272-274: Please be more cautious in these statements. While results are interesting, they do not show that an increased embryonic movement rate has adaptive benefits for brood parasites.

Lines 313-314: "Embryo movement is reliably detected in altricial embryos from approximately 25% into the incubation" This sentence would need to be reworded/clarified.

Line 315: "the same incubation stage". Considering that the study includes different bird species - some of them relatively distant phylogenetically -, the concept of "incubation stage" should be used carefully when comparing different species. I'm just saying that it should be carefully explained in the main text (not just in the supplementary information), especially considering that it is a key concept in this study.

Lines 319-320: "heart rate". Was heart rate not recorded or not included? Please, make it clear if heart rate and EMR were measured separately. It should be clearly stated if heart rate was decided not to include, or if data on heart rate are being neglected because muscle movement prevents to quantify it.

Lines 327-328: "If no movement was detected after this, no measurement was made". Why? Please clarify it.

Line 344: Why was pseudoreplication avoided considering only host eggs? Parasitic and host eggs in the same nest are also non-independent.

Line 349: "or other naturally occurring factors". Such as which? Please specify such "naturally occurring factors" or remove this sentence.

Lines 391-394: Incubation stages should be described earlier (in "General field methods") and in much more detail.

Author's Response to Decision Letter for (RSPB-2021-1137.R0)

See Appendix A.

RSPB-2021-1137.R1 (Revision)

Review form: Reviewer 1

Recommendation

Accept as is

Scientific importance: Is the manuscript an original and important contribution to its field?

Excellent

General interest: Is the paper of sufficient general interest?

Good

Quality of the paper: Is the overall quality of the paper suitable?

Excellent

Is the length of the paper justified?

Yes

Should the paper be seen by a specialist statistical reviewer?

No

Do you have any concerns about statistical analyses in this paper? If so, please specify them explicitly in your report.

No

It is a condition of publication that authors make their supporting data, code and materials available - either as supplementary material or hosted in an external repository. Please rate, if applicable, the supporting data on the following criteria.

Is it accessible?

Yes

Is it clear?

Yes

Is it adequate?

Yes

Do you have any ethical concerns with this paper?

No

Comments to the Author

I appreciate your thorough revision.

Review form: Reviewer 2

Recommendation

Major revision is needed (please make suggestions in comments)

Scientific importance: Is the manuscript an original and important contribution to its field?

Good

General interest: Is the paper of sufficient general interest?

Good

Quality of the paper: Is the overall quality of the paper suitable?

Good

Is the length of the paper justified?

Yes

Should the paper be seen by a specialist statistical reviewer?

No

Do you have any concerns about statistical analyses in this paper? If so, please specify them explicitly in your report.

No

It is a condition of publication that authors make their supporting data, code and materials available - either as supplementary material or hosted in an external repository. Please rate, if applicable, the supporting data on the following criteria.

Is it accessible?

Yes

Is it clear?

Yes

Is it adequate?

Yes

Do you have any ethical concerns with this paper?

No

Comments to the Author

The answers provided by the authors are clear, and most comments from the previous review have been conveniently addressed. This new submission is an improved version of the previous manuscript on the increased embryo movement found in avian brood parasites. I appreciate the detailed response provided by the authors, and I agree these results are novel and invite future research opportunities; however, I still believe that some evidence on the adaptive value of increased embryonic movement would need to be provided to interpret the results within the framework of the proposed hypothesis: "that increased embryonic movement assists avian brood parasites to achieve the necessary muscular and skeletal development needed for both the tasks of hatching from thicker eggshells and, in highly virulent species, killing or outcompeting their nestmates" (lines 93-96). Otherwise, it is difficult to discard non-adaptive explanations (even if authors consider them less plausible): bird embryos are known to alter their motility when exposed to different environmental factors - is there any chance that increased embryonic movement in avian brood parasites is simply a by-product of being in the nest (interacting) of other bird species (and thus increasing over the course of incubation)? In my opinion, the adaptive value of the studied trait (increased embryonic movement) is crucial to interpret the relevance of the results. Perhaps a more detailed discussion of alternative non-adaptive explanations would be helpful.

Lines 351-352: "in two species of these species". Please provide the name of these species.

Decision letter (RSPB-2021-1137.R1)

27-Sep-2021

Dear Ms McClelland

I am pleased to inform you that your manuscript RSPB-2021-1137.R1 entitled "Embryo movement is more frequent in avian brood parasites than birds with other reproductive strategies" has been accepted for publication in Proceedings B.

The referee(s) have recommended publication, but also suggest some minor revisions to your manuscript. Therefore, I invite you to respond to the referee(s)' comments and revise your manuscript. Because the schedule for publication is very tight, it is a condition of publication that you submit the revised version of your manuscript within 7 days. If you do not think you will be able to meet this date please let us know.

[http://datadryad.org/submit?journalID=RSPB&manu=\(Document not available\)](http://datadryad.org/submit?journalID=RSPB&manu=(Document%20not%20available)) which will take you to your unique entry in the Dryad repository. If you have already submitted your data to dryad you can make any necessary revisions to your dataset by following the above link. Please see <https://royalsocietypublishing.org/journals/ethics-policies/data-sharing-mining/> for more details.

6) For more information on our Licence to Publish, Open Access, Cover images and Media summaries, please visit <https://royalsocietypublishing.org/journals/authors/author-guidelines/>.

Sincerely,

Dr Sasha Dall

<mailto:proceedingsb@royalsocietypublishing.org>

Associate Editor:

Board Member: 1

Comments to Author:

The authors have handled these revisions well and with care. I note some minor typos, listed below, and suggest that the authors also provide species names in the lines requested by reviewer 2. My view is that the study will be an excellent contribution that will spark interest and further study of this topic.

Line 72: change the second 'which' to 'that'

Line 449: add 'are' before 'available'

Line 452: delete 'the frontend'

Line 488: Change 'which' to 'that'

Line 498: it should read 'egg and nest identity were included as random effects'

Reviewer(s)' Comments to Author:

Referee: 1

Comments to the Author(s)

I appreciate your thorough revision.

Referee: 2

Comments to the Author(s)

The answers provided by the authors are clear, and most comments from the previous review have been conveniently addressed. This new submission is an improved version of the previous manuscript on the increased embryo movement found in avian brood parasites. I appreciate the detailed response provided by the authors, and I agree these results are novel and invite future research opportunities; however, I still believe that some evidence on the adaptive value of increased embryonic movement would need to be provided to interpret the results within the framework of the proposed hypothesis: "that increased embryonic movement assists avian brood parasites to achieve the necessary muscular and skeletal development needed for both the tasks of hatching from thicker eggshells and, in highly virulent species, killing or outcompeting their nestmates" (lines 93-96). Otherwise, it is difficult to discard non-adaptive explanations (even if authors consider them less plausible): bird embryos are known to alter their motility when exposed to different environmental factors - is there any chance that increased embryonic movement in avian brood parasites is simply a by-product of being in the nest (interacting) of other bird species (and thus increasing over the course of incubation)? In my opinion, the adaptive value of the studied trait (increased embryonic movement) is crucial to interpret the

relevance of the results. Perhaps a more detailed discussion of alternative non-adaptive explanations would be helpful.

Lines 351-352: "in two species of these species". Please provide the name of these species.

Author's Response to Decision Letter for (RSPB-2021-1137.R1)

See Appendix B.

Decision letter (RSPB-2021-1137.R2)

04-Oct-2021

Dear Ms McClelland

I am pleased to inform you that your manuscript entitled "Embryo movement is more frequent in avian brood parasites than birds with other reproductive strategies" has been accepted for publication in Proceedings B.

Data Accessibility section

Open Access

Paper charges

Sincerely,
Editor, Proceedings B
<mailto:proceedingsb@royalsociety.org>

Appendix A

Proceedings of the Royal Society B

RE: RSPB-2021-1137

Dear Proceedings of the Royal Society B Editorial Team,

We very much appreciate the time that the reviewers, Board Member and yourselves have taken in reading and commenting on our manuscript submission to *Proceedings of the Royal Society B* - '**Embryo movement is more frequent in avian brood parasites than birds with other reproductive strategies**', and for providing us with the opportunity to revise our paper. Please find below our responses (in **bold**) to the reviewer's comments (in *italics*). We thank the reviewers for their incredibly useful, positive and insightful feedback, and the overall constructive nature of their reviews, both of which have greatly improved our paper.

We look forward to hearing from you in due course.

Yours faithfully,

Stephanie McClelland and Co-authors

Associate Editor

I have received two expert reviews of the manuscript. Both reviewers and I found this to be a very interesting study, tackling an interesting research question in an innovative way. The study uses phylogenetically-controlled analyses to compare embryonic movement in brood-parasitic and non-parasitic birds, and finds that brood-parasitic species have increased movement throughout embryonic development, suggesting convergent evolution. Both reviewers provide a list of suggestions that I hope the authors will find helpful in improving the manuscript. Reviewer 2 raises an important point about non-independence of parasitic and host eggs, and this needs addressing. Both reviewers point out that evidence for the benefits of embryonic movement in post-embryonic life is not provided, and I agree that additional discussion of this would strengthen the paper. Both reviewers also ask for clarification of the methods.

Thank you for your positive feedback on this manuscript and we are glad that you found it interesting. We have addressed the comments and concerns of the reviewers below, and we are grateful for their useful advice and comments.

Referee: 1

This is an exciting study that explores the idea that embryo movement in birds is shaped by selection. Using a phylogenetically controlled analysis, the authors find a pattern of more embryo movement in brood parasites than their hosts. The interpretation is that higher embryonic movement rate in brood parasites evolved convergently between lineages because a) brood parasites engage in activities that require strength (breaking out of thick egg shell, evicting host hatchlings), and b) brood parasites with a stronger muscu-skeletal system would be more successful at both activities. Embryo movement is predicted to strengthen muscular development, though this was not tested. Your description of data scoring from the buddy egg monitor needs clarification and I suggest a different reference for its use in birds.

Thank you for your comments. We are glad that you found the study to be exciting and we appreciate your useful suggestions for revisions. We have addressed your specific concerns below, which have further helped to clarify the methods and scope of this study.

Comments

Line 157: "This suggesting" is awkward

Thank you; this sentence has been modified to improve the grammar:

Lines 159-160: "This suggests that the demands of hatching and virulence in common cuckoos may have driven their relatively high EMR."

Table 1: you show that little bee-eaters have a lot of embryo movement; somewhere in the manuscript text or table you may want to indicate that little bee eaters excavate sand tunnels. Therefore, embryo movement in general could be selected for in systems with high activity level, such as excavation behaviour. Perhaps such a thought could be included as a footnote for this table as the embryo movement pattern in little bee-eaters is strikingly different from that recorded in other non-brood parasites.

Thanks very much indeed for this comment. This is a really interesting idea to explain the high rate of movement in little bee-eaters. However, these birds would not be involved in tunnel excavation until adulthood and additionally another species measured (black-collared barbets) is also a primary nest excavator and does not show this greater movement rate. As such we feel there is insufficient evidence to make this suggestion. We do not have an alternative explanation for why these species exhibit particularly high movement rates however, and we have included acknowledgement of this to the results and discussion.

Line 166-170: "Unlike the lesser honeyguides and their hosts, the slope of increase of EMR in greater honeyguides did not differ significantly from that of their hosts, little bee-

eaters (Merops pusillus) (slope \pm SE = 2.91 \pm 7.67, t_{191} = 0.38, p = 0.70, Fig 3b), which themselves had a relatively high EMR.”

Lines 268-270 : “Additionally, little bee-eaters, hosts of the greater honeyguides, exhibited relatively high rates of embryo movement compared to other non-parasite species, a curious finding for which we currently do not have an explanation. .”

Lines 249-270: Your explanations are convincing; however, see comment above regarding behavioural ecology of the host, which you neglect in this paragraph.

Please see our above response regarding this.

Line 269: embryonic movement does not evolve. This needs to be rewritten.

We have adjusted the wording of this sentence:

Line 265 - 268: “That these two honeyguide species are congeneric [34] makes this difference particularly striking, as it suggests that embryonic behaviour has the potential to evolve rapidly in response to differences in host behaviour and morphology.”

Line 275: change “with” to “that”

Thank you; we have corrected this error:

Line 274-275: “The factors influencing variation in embryonic movement across incubation are less well understood, as are the mechanisms that control this movement.”

Line 285: change from “rate is driven primarily by ecology rather than common ancestry” to “rate is driven primarily by intrinsic or environmental factors rather than common ancestry”

Thank you for your suggestion. We have made this change:

Line 282-285: “Phylogenetic position did not significantly predict a species’ embryonic movement rate, suggesting that variation in embryonic movement rate is driven primarily by intrinsic or environmental factors rather than common ancestry;”

Discussion in general: you do not use the opportunity to make predictions for future research. For example, the magnitude of embryonic movement could predict time to evict host and a range of other measurable parameters.

We appreciate this suggestion and have included more future research suggestions to our discussion:

Line 297-304: “Future research could directly measure the consequences of greater embryonic movement rates on the muscle density and performance of individuals of these parasitic species. For example, if this embryonic trait has an adaptive benefit for brood parasites, then we would expect increased embryo movement to increase the parasitic chick’s efficiency at evicting or killing host offspring. Additionally, it would be informative to measure embryo movement in other species that experience challenging nestling social environments, such as nest-sharing colonial breeders or species with large asynchronous clutches and/or high rates of siblicide [42,43].”

Line 309: replace reference 44 with Colombelli-Négrel, D., Hauber, M.E. and Kleindorfer, S., 2014. Prenatal learning in an Australian songbird: habituation and individual discrimination in superb fairy-wren embryos. Proceedings of the Royal Society B: Biological Sciences, 281(1797), p.20141154.

Colombelli-Négrel et al (2014) were the first to use the “Egg Buddy” for avian research, the same model you also used – previous research in birds used acoustocardiography. The study you cite, reference 44 published in 2018, followed on from Colombelli-Négrel et al (2014).

Thank you for highlighting this. We have added this reference accordingly as reference number 47:

Line 314-315: “[...]and it has been used to monitor embryo development and heart rate in both birds and reptiles [45,46,47].

Line 318: the heart rate measure does NOT become unreliable; rather, there is more movement. The egg buddy clearly shows the difference between heart rate bpm and movement on the screen with different symbols (heart symbol, moving bird symbol). You could still have recorded/analysed heart rate but you would need to restrict the notation of heart rate to the seconds without embryo movement.

We appreciate your correction on this, and have adjusted the description of this in our methods. However, we did not record heartrate in this study. This would have been difficult due to the high frequency of movement occurring for most species.

Line 324-326: “However, as the size and activity of the embryo increases, the heart rate measure becomes more challenging to record due to the increased muscular movements of the embryo [42].”

Line 328: you write “embryo movement was displayed in real time on the screen of the egg monitor” and then line 330: “The number of embryo movements were counted from playback of the video recording”. This is confusing and needs to be rewritten to reflect the fact that the buddy egg monitor screen has a symbol showing embryo movement per second; the display duration on the screen is 1 sec for both heart rate as beats per minute and for any movement event even if the event was less than 1 sec; so what you can score is the number of seconds that had embryo movement. Also, to avoid confusion with the term “playback the video” (because a common research approach is to use playback to elicit a behavioural response) you could state that you watched the video and scored the number of seconds with embryo movement events.

We have rewritten this section of the methods for clarification. Our methods for quantifying embryo movement followed Pollard et al.’s 2016 study. This study validated the use of the ‘EggBuddy’ by showing that counting movements of the embryo as changes in the configuration of the bird icon gave significantly comparable result to visually counting movement by ‘windowing’ the egg (a more invasive method). While the “heartrate per minute” is displayed each second on the screen, the movements are displayed as individual movements in a continuous data stream and the screen display changes significantly faster than once per second, requiring the video recording to be slowed down to count each movement as it occurs.

Reference: Pollard AS, Pitsillides AA, Portugal SJ. *Validating a Noninvasive Technique for Monitoring Embryo Movement In Ovo*. *Physiol Biochem Zool*. 2016;89(4):331–9

Lines 334-339: “Embryo movement was displayed in real-time on the screen of the egg monitor as an animated bird symbol which changed configuration when movement was detected. A 60-second video of the screen was recorded immediately following acclimation, and subsequently analysed. The number of embryo movements was counted from watching the video recordings at 0.5 x speed. This was performed by either SCM, MC, or MR, and blindly to the specimen ID .”

Lines 344-346: you write that you sampled one host egg per nest, and each egg was out of the nest for max 10 minutes. This does not need to be repeated elsewhere.

Thank you for pointing this out. We have fixed this.

Line 372: this contradicts one host egg per nest when you write “along with any live host eggs present”

We have corrected this statement:

Line 381-383: “Embryo movement measurements were taken from the parasite egg in each nest located, along with a live host egg if present.”

Line 410: why did you return hatched cuckoo chicks to unparasitised host nests? You collected the abandoned cuckoo egg from the wild; why would you return an abandoned cuckoo egg that you reared in a lab to a new host nest?

Hatched cuckoo chicks were returned to the wild by placement in unparasitized nests as part of an unrelated study. The return of cuckoo chicks following egg manipulations is standard practise at the Prof. Honza’s fieldsite in the Czech Republic. Common cuckoo populations are decreasing across Europe, by as much as 69% in some regions, and as such returning these chicks to the wild was sensible for the population (Denerley et al., 2019).

Line 419-420: repeats the info about one host egg per nest

Thank you for highlighting this; fixed.

Lines 442-443: the egg monitor records seconds with egg movement; this needs to be rewritten. Perhaps you included seconds with both ‘bird movement symbol’ and ‘zero’ as movement. If so, this needs to be stated. A bird movement symbol would never be a false zero. A zero would indicate that no signal could be detected (movement or heart rate). I agree that intense movement could result in a zero signal, and this is unlikely in very young embryos.

Recordings that displayed the bird icon but did not show any movement of the icon (change of configuration from ‘wings up’ to ‘wings down’) were considered as zero embryo movement. However, we did not record the heartrate on any recordings, and were unable to validate whether these zeros were an accurate recording of no movement, or whether a signal could not be obtained due to the small embryo size, and as such excluded these measurements.

Referee: 2

This study was aimed to test the hypothesis that (lines 92-95) increased movement during the embryonic life would allow avian brood parasites to achieve the necessary muscular development to successfully exploit their hosts (e.g., hatching from thicker eggshells, killing or out-competing their non-parasitic nestmates). The authors found that embryos of some avian brood parasites exhibit increased muscular movement compared to some of their main host species, and some closely-related non-parasitic species. On the other hand, there seems to be no link between embryonic movement rate and parasite virulence. I enjoyed reading the manuscript. Overall, this is an interesting and well-written manuscript that provide novel results on a little explored aspect of avian brood parasitism. However, I have some concerns regarding the main hypothesis of the study and the interpretation of results.

Thank you for your comments. We are happy to hear that you appreciated the paper and found it interesting, novel and well-written. We appreciate your suggestions to improve the manuscript and have incorporated and addressed them, as described below.

1) The main hypothesis of the study focuses on the potential adaptive benefits that increased embryonic movement would have for brood parasitic species (this idea is present throughout the entire manuscript); however, the authors do not provide any evidence of such benefits (e.g., correlation between embryonic movement rate (EMR) and brood parasite's muscle development; a link between EMR and hatching success or competitive abilities). I know it is difficult to fix at this point, but author should provide some direct evidence on the competitive advantage (benefit) that increased embryonic movement would provide for brood parasites. Otherwise, one should be more cautious when referring to the increased EMR as "embryonic adaptation".

We appreciate that this is a limitation of this study. Unfortunately, directly measuring the effect of increased embryo movement on the musculature in these particular species is outside the scope of this study and would require more invasive procedures that would not be possible due to the protected/vulnerable status of many of these species. Similarly, getting these species to breed in captivity is also incredibly difficult, if not impossible for many of them, making it incredibly difficult to conduct such studies.

However, the adaptive role of embryo movement in determining early life physiology, particularly in specializing the musculoskeletal structure for specific postnatal mechanical roles, is already strongly supported in other avian species (Heywood et al., 2005; Pitsillides, 2006; Pollard et al., 2014, 2017).

Previous work on our focal species also provides ample evidence of behavioral traits and musculature adaptations that are not exhibited by non-parasitic species. Parasitic common cuckoos, for example, have an increased density of primary muscle fibers in their musculus complexus, the neck muscles which is involved in hatching (Honza et al., 2015). This muscle complex is also important for begging behaviour, which regulates dorsal flexion of the neck and coordination of head movement, and thereby effects competitive interactions between nestlings (Lipar and Ketterson, 2000). This suggests it could likewise explain prolonged begging stamina seen in brown-headed cowbirds (*Molothrus ater*) (Dearborn et al., 2009; Soler et al.,

1999). Combining this information with findings that embryo hyperactivity increases primary muscle fiber density in chickens (*Gallus gallus*) and also consequently effects postnatal strength (Pitsillides, 2006), provides a plausible link between brood parasite physiology and embryo movement. While unfortunately there is little published information on the musculoskeletal system of many brood-parasitic hatchlings, the behavior of many of these species shows evidence of increased strength very shortly after hatching, such as the considerable bite force and shaking behavior of honeyguides (Spottiswoode and Koorevaar, 2012) and the eviction of nestmates by all members of *Cuculus*, *Cacomantis* and *Chrysococcyx* cuckoo species (Honza et al., 2007). Our data on the greater increase in embryo movement in these parasitic species (compared to non-parasitic species) are correlational and therefore cannot conclusively show that embryo movement is the mechanism enabling this strength, but does provide evidence of a highly plausible explanation for something that is previously unknown and unexplored. We therefore consider that our results are novel and invite future research opportunities. However, we appreciate your concern regarding our wording when we discuss how embryo movement could be adaptative for these species, and as such we have adjusted the discussion to make it clearer that we are highlighting this trait as a potential adaptation. Please see wording changes in response to comments below.

2) Similarly, some predictions would need to be supported by stronger evidence. For example, the authors should be more cautious in speculating on the muscular demands, and the degree of musculoskeletal development, required by the different parasitic strategies (lines 121-126). Why are virulent brood parasites (those that eliminate all host eggs or kill the host offspring) expected to have higher muscular demands compared to less virulent brood parasites (those that share the nest with the parasitic offspring and, therefore, have to compete with their nestmates)? Is there any evidence to support that idea? If not, would ejector parasites be necessarily predicted to show increased embryonic movement rate than non-ejector parasites?

Our speculation of the muscular demands of virulent brood parasites is based on studies which have suggested that behaviour such as eviction is energetically costly to the hatchling (Anderson et al., 2009; Grim et al., 2009). Honeyguide species which kill host young by biting and shaking them appear to need a 'rest' after vigorous bouts of this behaviour and exhibit visible deep breathing after this activity (Spottiswoode and Koorevaar, 2012). However, we do not know of any study that has directly compared the effort of highly virulent parasitic chicks to that of low-virulence, nest-mate tolerant parasitic chicks. This prediction therefore relies on an assumption that virulent species experience greater muscular demands. We address this in our discussion (differences in strenuousness of virulence behaviour, lines 247-270) as a possible reason why we do not see significant differences between high and low virulent parasites.

3) Interpretation of results. This study shows that some avian brood parasites have a higher overall rate of increase in EMR over the course of incubation compared to non-parasitic species. While this is an interesting result, the authors should be cautious in stating that this is an "embryonic adaptation to a brood-parasitic lifestyle [...] explained through convergent evolution". To state that, they should provide evidence for the potential adaptive benefit of increased EMR. Do the authors have any data (e.g., hatching success, begging rate) to suggest that brood parasites

benefit from increased embryonic movement? In my opinion, this would make the difference and would improve the study substantially.

We appreciate your concern about the strength of our interpretation. We have clarified the statement you highlight to make it clear that we are proposing that this embryonic behaviour is *potentially* adaptive based on what is known about how avian muscle development is influenced by embryonic movements and hatchling behaviour of these species. We believe that the novel correlation we have uncovered between parasitism and embryo behaviour will stimulate new investigation into the fitness consequences of variation in embryo movement, since (as our study highlights) very little research has been done on the developmental physiology of avian brood parasites. Unfortunately, we do not yet have the suggested data (begging rate, etc.), but agree this would be a worthwhile future direction for continuing this research and have added it to our discussion:

Line 290-292: “[..] suggesting that this is potentially an embryonic adaptation to a brood-parasitic lifestyle [..]”

Line 297-304: “Future research could directly measure the consequences of greater embryonic movement rates on the muscle density and performance of individuals of these parasitic species. For example, if this embryonic trait has an adaptive benefit for brood parasites, then we would expect increased embryo movement to increase the parasitic chick’s efficiency at evicting or killing host offspring. Additionally, it would be informative to measure embryo movement in other species that experience challenging nestling social environments, such as nest-sharing colonial breeders or species with large asynchronous clutches and/or high rates of siblicide [42,43].”

Other comments:

Lines 272-274: Please be more cautious in these statements. While results are interesting, they do not show that an increased embryonic movement rate has adaptive benefits for brood parasites.

Please see our above response regarding how we aimed to discuss to possible adaptive function of embryo movement. The sentence you refer to here does not mention adaptive benefits (and as such we have not adjusted it), but rather that we propose that embryo movement may have evolved convergently in these species.

Lines 313-314: “Embryo movement is reliably detected in altricial embryos from approximately 25% into the incubation” This sentence would need to be reworded/clarified.

We have reworded this sentence:

Line 319-320: “Embryo movement is reliably detected in altricial embryos after approximately the first quarter of the incubation period”

Line 315: “the same incubation stage”. Considering that the study includes different bird species – some of them relatively distant phylogenetically –, the concept of “incubation stage” should be used carefully when comparing different species. I’m just saying that it should be carefully explained in the main text (not just in the supplementary information), especially considering that it is a key concept in this study.

Our estimation of embryo stage was based primarily on previously published systems which we reference in our general methods. However, we agree that a more detailed explanation is required. We think that this would be too long to include in the main manuscript. As it is secondary to our core methods we have included it in the supplementary information. We are happy to move this section into the main manuscript if the editor feels it is necessary, but meanwhile we have included a brief statement of the methods in the main manuscript, and clearer reference to the details available in the supplementary material:

Line 360-364: “Egg stage was estimated by visual examination of the embryo via candling and a stage of development assigned based on the embryo size and appearance, the albumen colouration, blood vessel quantity and air cell size. Further details and illustrations of embryo stages are available in the supplementary material.”

Lines 319-320: “heart rate”. Was heart rate not recorded or not included? Please, make it clear if heart rate and EMR were measured separately. It should be clearly stated if heart rate was decided not to include, or if data on heart rate are being neglected because muscle movement prevents to quantify it.

Heart rate was not recorded in this study. With this method, measurements of heart rate can't be accurately recorded when frequent embryo movement is occurring, which given the high movement frequency in these species would have made this data very difficult to collect.

Lines 324-326: “However, as the size and activity of the embryo increases, the heart rate measure becomes more challenging to record due to the increased muscular movements of the embryo [42]. Therefore, we did not record heart rate.”

Lines 327-328: *"If no movement was detected after this, no measurement was made". Why? Please clarify it.*

From personal observation, if movement wasn't detected immediately it was unlikely to be picked up and would suggest the egg is infertile, dead or too early in development. In order not to prolong disturbance at the nest, we did not wait longer to detect a signal, as some species were likely to abandon nests if they were disturbed for too long. We have clarified this in the manuscript:

Lines 333-334: *"If no movement was detected after this, no measurement was made to minimize disturbance."*

Line 344: *Why was pseudoreplication avoided considering only host eggs? Parasitic and host eggs in the same nest are also non-independent.*

We appreciate this suggestion and recognise it is a valid consideration. We therefore adjusted the model to include 'nest identity' as a random effect to account for this non-independence. This did not change the results significantly, as can be seen from the updated Results section below. Additionally, when we addressed this, we realised there was a minor error in our description of our methods. While in most cases we avoided sampling more than one host egg from the same nest, for a couple of species (zitting cisticolas and little bee-eaters) due to low nest availability we did take measurements of two host eggs from a nest. However, we have now accounted for this statistically with the inclusion of nest identity in the models and, as stated above, we found that this does not significantly influence our results or conclusions. We apologise for this oversight in our original submission and have rectified the relevant sections of the methods and results and supplementary dataset.

See changes to the statistical methods and results:

Line 349-353: *"Where possible we measured only a single egg per host clutch to avoid pseudoreplication since host eggs in the same clutch are non-independent. However, due to limited nest availability, in some cases two host eggs were sampled from the same clutch. This was accounted for statistically by including nest identity as a random factor in all analyses."*

Line 505-506: *"Similarly, nest identity was included as a random variable to account for eggs which were sampled from the same host nest. The model with the best fit for predicting EMR in*

these species included fresh egg mass and the interaction of parasitic status and incubation stage as fixed factors, and egg identity and nest identity as random factors.”

Line 349: “or other naturally occurring factors”. Such as which? Please specify such “naturally occurring factors” or remove this sentence.

As suggested, we have modified this sentence. As these data were collected from multiple field sites with variable conditions, nests were occasionally lost or dropped from the study for a wide variety of reasons, such as being trampled by cattle, or flooded. We feel it would be excessive to list all of these for each egg/nest loss, and as such have grouped them as ‘nest destruction’, to be clear.

Line 356-357: “Repeat measures were not obtained for some eggs due to clutch loss from predation, nest destruction or host rejection.”

Lines 391-394: Incubation stages should be described earlier (in “General field methods”) and in much more detail.

See our above response.

References:

- Anderson, M.G., Moskát, C., Bán, M.S., Grim, T., Cassey, P., Hauber, M.E., and Iwaniuk, A. (2009). Egg Eviction Imposes a Recoverable Cost of Virulence in Chicks of a Brood Parasite. *PLoS One* 4, e7725.
- Dearborn, D.C., MacDade, L.S., Robinson, S., Dowling Fink, A.D., and Fink, M.L. (2009). Offspring development mode and the evolution of brood parasitism. *Behav. Ecol.* 20, 517–524.
- Denerley, C., Redpath, S.M., Wal, R. van der, Newson, S.E., Chapman, J.W., and Wilson, J.D. (2019). Breeding ground correlates of the distribution and decline of the Common Cuckoo *Cuculus canorus* at two spatial scales. *Ibis (Lond. 1859)*. 161, 346–358.
- Grim, T., Rutila, J., Cassey, P., and Hauber, M.E. (2009). The cost of virulence: An experimental study of egg eviction by brood parasitic chicks. *Behav. Ecol.* 20, 1138–1146.
- Heywood, J.L.L., Mcentee, G.M.M., and Stickland, N.C.C. (2005). In ovo neuromuscular stimulation alters the skeletal muscle phenotype of the chick. *J. Muscle Res. Cell Motil.* 26, 49–56.
- Honza, M., Vošlajerová, K., and Moskát, C. (2007). Eviction behaviour of the common cuckoo *Cuculus canorus* chicks. *Commun. J. Avian Biol* 38, 385–389.

- Honza, M., Feikusová, K., Procházka, P., and Picman, J. (2015). How to hatch from the Common Cuckoo (*Cuculus canorus*) egg: implications of strong eggshells for the hatching muscle (*musculus complexus*). *J. Ornithol.* 156, 679–685.
- Lipar, J.L., and Ketterson, E.D. (2000). Maternally derived yolk testosterone enhances the development of the hatching muscle in the red-winged blackbird *Agelaius phoeniceus*. *Proc. R. Soc. B Biol. Sci.* 267, 2005–2010.
- Pitsillides, A.A. (2006). Early effects of embryonic movement: “a shot out of the dark.” *J. Anat.* 208, 417–431.
- Pollard, A.S., McGonnell, I.M., and Pitsillides, A.A. (2014). Mechanoadaptation of developing limbs: shaking a leg. *J. Anat.* 224, 615–623.
- Pollard, A.S., Boyd, S., McGonnell, I.M., and Pitsillides, A.A. (2017). The role of embryo movement in the development of the furcula. *J. Anat.* 230, 435–443.
- Soler, M., Soler, J.J., Martínez, J.G., and Moreno, J. (1999). Begging behaviour and its energetic cost in great spotted cuckoo and magpie host chicks. *Can. J. Zool.* 77, 1794–1800.
- Spottiswoode, C.N., and Koorevaar, J. (2012). A stab in the dark: chick killing by brood parasitic honeyguides. *Biol. Lett.* 8, 241–244.

Appendix B

Proceedings of the Royal Society B

RE: RSPB-2021-1137.R1

Dear Proceedings of the Royal Society B Editorial Team,

We are delighted that this paper has been accepted for publication in Proceedings B. We very much appreciate the time that the reviewers, Board Member and yourselves have taken in reading and commenting on our manuscript. Please find below our responses (in **bold**) to the reviewer's and editors comments (in *italics*). We thank them for their incredibly useful, positive, and insightful feedback.

Yours faithfully,
Stephanie McClelland and Co-authors

Associate Editor:

Board Member: 1

Comments to Author:

The authors have handled these revisions well and with care. I note some minor typos, listed below, and suggest that the authors also provide species names in the lines requested by reviewer 2. My view is that the study will be an excellent contribution that will spark interest and further study of this topic.

Line 72: change the second 'which' to 'that'

Line 449: add 'are' before 'available'

Line 452: delete 'the frontend'

Line 488: Change 'which' to 'that'

Line 498: it should read 'egg and nest identity were included as random effects'

Thank you for pointing these out, we have corrected these issues in the manuscript. Please see track changes in the revised version.

Reviewer(s)' Comments to Author:

Referee: 1

Comments to the Author(s)

I appreciate your thorough revision.

Thank you for your helpful suggestions during the review process.

Referee: 2

Comments to the Author(s)

The answers provided by the authors are clear, and most comments from the previous review have been conveniently addressed. This new submission is an improved version of the previous manuscript on the increased embryo movement found in avian brood parasites. I appreciate the detailed response provided by the authors, and I agree these results are novel

and invite future research opportunities; however, I still believe that some evidence on the adaptive value of increased embryonic movement would need to be provided to interpret the results within the framework of the proposed hypothesis: "that increased embryonic movement assists avian brood parasites to achieve the necessary muscular and skeletal development needed for both the tasks of hatching from thicker eggshells and, in highly virulent species, killing or outcompeting their nestmates" (lines 93-96). Otherwise, it is difficult to discard non-adaptive explanations (even if authors consider them less plausible): bird embryos are known to alter their motility when exposed to different environmental factors - is there any chance that increased embryonic movement in avian brood parasites is simply a by-product of being in the nest (interacting) of other bird species (and thus increasing over the course of incubation)? In my opinion, the adaptive value of the studied trait (increased embryonic movement) is crucial to interpret the relevance of the results. Perhaps a more detailed discussion of alternative non-adaptive explanations would be helpful.

We are pleased to hear that you approved of the revised manuscript, and we are grateful for your comments. We appreciate your perspective on how we discuss the potential adaptive value of this trait and we have tried to be clear in our work about the limitation of this data to conclusively demonstrate the adaptive function of embryo movement. On your suggestion, we have added further to this in the discussion.

Line 297-306: "While the evidence is consistent with our theory of an adaptive function of embryo movement, this cannot be conclusively determined with this data, and it is possible that other aspects of a parasitic lifestyle could induce or select for greater movement. Future research could address this by directly measure the consequences of greater embryonic movement rates on the muscle density and performance of individuals of these parasitic species. For example, if this embryonic trait has an adaptive benefit for brood parasites, then we would expect increased embryo movement to increase the parasitic chick's efficiency at evicting or killing host offspring."

Lines 351-352: "in two species of these species". Please provide the name of these species.

Thank you, we have amended this for clarity.

Lines 351-352 "However, due to limited nest availability of zitting cisticolas and little bee-eaters, two eggs were sampled from the same clutch."